# Microfluidic Preconcentration Chip with Self-Assembled Chemical Modified Surface for Trace Carbonyl Compounds Detection

**DOI:** 10.3390/s18124402

**Published:** 2018-12-13

**Authors:** Jie Cheng, Jianwei Shao, Yifei Ye, Yang Zhao, Chengjun Huang, Li Wang, Mingxiao Li

**Affiliations:** 1R&D Center of HealthCare Electronics, Institute of Microelectronics of Chinese Academy of Sciences, Beijing 100029, China; chengjie@ime.ac.cn (J.C.); yeyifei@ime.ac.cn (Y.Y.); zhaoyang@ime.ac.cn (Y.Z.); 2School of Future Technology, University of Chinese Academy of Sciences, Beijing 101400, China; 3The State Key Laboratory of Chemical Resource Engineering, Beijing 100029, China; jwshao1993@163.com; 4College of Materials Science and Engineering, Beijing University of Chemical Technology, Beijing 100029, China

**Keywords:** click chemistry, surface modification, trace carbonyl compounds, micro columnar preconcentrator, microfluidic

## Abstract

Carbonyl compounds in water sources are typical characteristic pollutants, which are important indicators in the health risk assessment of water quality. Commonly used analytical chemistry methods face issues such as complex operations, low sensitivity, and long analysis times. Here, we report a silicon microfluidic device based on click chemical surface modification that was engineered to achieve rapid, convenient and efficient capture of trace level carbonyl compounds in liquid solvent. The micro pillar arrays of the chip and microfluidic channels were designed under the basis of finite element (FEM) analysis and fabricated by the microelectromechanical systems (MEMS) technique. The surface of the micropillars was sputtered with precious metal silver and functionalized with the organic substance amino-oxy dodecane thiol (ADT) by self-assembly for capturing trace carbonyl compounds. The detection of ppb level fluorescent carbonyl compounds demonstrates that the strategy proposed in this work shows great potential for rapid water quality testing and for other samples with trace carbonyl compounds.

## 1. Introduction

Trace organic pollutants in water pose a threat to the quality of drinking water. The carbonyl compounds in water are easily released into the atmosphere or transferred to food directly or indirectly to cause a variety of diseases. For example, aldehyde compounds have strong stimulating effects on the human eyes, skin, and respiratory tract [1,2]. Some carboxylic acids and aldehydes, such as formaldehyde and acetaldehyde have also been identified as carcinogens or suspected carcinogens. Carbonyl compounds have received widespread attention due to their harm to humans [3,4]. Environmental water samples contain a complex mixture of carbonyl compounds at very low concentrations typically in the parts per billion (ppb) range [5], which are difficult to detect. Conventional methods of trace-level measurement are based on laboratory off-line instrument analysis, including gas chromatography (GC), high performance liquid chromatography (HPLC), mass spectrometry (MS), infrared spectroscopy (IR), cavity ring-down spectroscopy (CDRS) as well as GC–MS, HPLC–MS and other methods [6,7,8,9,10,11,12,13]. All of the above methods have the advantages of high sensitivity and strong selectivity, but the disadvantages of complicated sample preparation, long processing period, and expensive equipment purchase and maintenance are also obvious. Therefore, it is necessary to find a device for rapid in-the-field detection to provide technical support for effective prevention and control of sudden pollution events.

Recently, high performance and miniaturization of sample pretreatment structure and new detection methods based on chemical reactions became the focus of research. For chemical detection methods, detection methods based on chemical reaction have been proposed currently, in which chemical resistance sensor array [14,15,16] and a chemically modified surface for capturing molecules [17,18,19,20] can achieve rapid qualitative analysis and high sensitivity. The synthesis method based on click chemistry has been widely used in the surface functionalization of materials, functional polymer synthesis, dendrimer synthesis, cell markers, DNA markers etc. [21,22,23,24,25,26]. The synthesis method has characteristics of high yield, few by-products, high purity, and easy separation. The sensor prepared by this method has higher selectivity and stability than the traditional fixed marking method, also with simpler steps and improved utility. However, to date there are no suitable device structures for in-the-field detection combined with the click chemistry synthesis method for the detection of carbonyl compounds. Rapid development of MEMS technology provides a new means of specific molecular detection [27,28]. In particular, the micro total analysis system (μTAS) provides a portable integrated device for detection [17]. Various proposed devices integrate advanced sensors into lab-on-a-chip systems, thus introducing the advantages of microfluidics, such short analysis time, high sensitivity, and small sample consumption to the chemical assay application field [29,30,31]. Yeh et al. [32] developed a paper-based microfluidic device for sample pre-concentration with high enrichment factors. The Landers group published a series of papers focusing on using microchip-based solid-phase extract (SPE) for the purification of DNA [33,34]. Therefore, it is meaningful to design a chemical microfludic device suitable for rapid in-the-field carbonyl compounds detection.

In this work, we propose a silicon-based micro columnar preconcentrator (μCPC) using MEMS technology with a self-assembled chemically modified surface for capturing of trace carbonyl compounds. The micro pillars increase the surface area of the reaction and concentrate the sample, and are suitable for the detection of trace substances. Synthesis of amino-oxy dodecane thiol (ADT) based on the concept of click chemistry was implemented to modify the silicon-based surface. Carbonyl compounds are captured by surface modified groups through an oximation reaction. The proposed embedded package structure of the device adopts the microfluidic operation mode which can realize rapid detection. Compared to the reported physical adsorption device [27,28] and chemical sensor [14,15,20], our device has the advantages of easy operation, rapid in-the-field detection, and a trace level detection limit. The experiment results in our work verified that this surface modified μCPC structure has great potential for measuring trace carbonyl pollutants in water.

## 2. Materials and Methods

### 2.1. Materials

Polydimethylsiloxane (PDMS) prepolymer and a curing agent kit (SILPOT 184) were purchased from TORAY (Tokyo, Japan). Other reagents and solvents, including 3-perylenecarboxaldehyde (>95%) were purchased from Tokyo Chemical Industry Co. (Tokyo, Japan), dichloromethane (A.R.) and acetone (A.R.) were purchased from Beijing Chemical Works. The aminooxy-based reactive coating (ADT) was synthesized according to a published method [35]. Instruments used in the experiment include a scanning electron microscope (Phenom ProX), a syringe pump (Harvard Apparatus Pump 11 Elite), and a microscope (CX41 OLYMPUS).

### 2.2. μCPC Chip Design and Fabrication

The proposed microfluidic device is illustrated in Figure 1. The columnar structure preconcentrator was designed to achieve the enrichment of the sample by an increase of the contact area and a decline of fluid rate. Bifurcated channels were adopted at the sample inlet and outlet to realize the uniform flow and distribution of the sample in the micro columns effectively. The flow velocity distribution of the fluid under different micro column parameters was simulated by finite element simulation software COMSOL, the most suitable structural parameters were obtained, and then the layout and subsequent process were processed based on the data support.

The laminar flow module was used to simulate the liquid flow in the preconcentrator and guide the design of the preconcentrator structure in order to achieve uniform fluidic flow distribution in the gap of micropillars and a theoretical reference to the proper spacing of the micro pillars. The vertical flow velocities of the micropillars are uniform under steady state simulation, thus, the 3D models are simplified into 2D models for analysis. In the microfluidic channel, the flow velocity field is described by the Navier–Stokes equations [35,36]. The fluid is assumed to be Newtonian and incompressible. The simulation model is in 2D Cartesian coordinates and for a steady flow. The fluid properties such as the dynamic viscosity and the density are assumed to be constant. The material of the fluid is water with a dynamic viscosity of μ = 10^−3^ Pa s, and a fluid density of ρ = 1000 kg/m^3^, and the solid structural material is silicon based. For all simulations, the water inlet pressure was set to 10 kPa. Under the guidance of simulation data, the processing of the device is realized. The bifurcated input consists of six levels with one common inlet equally divided into two for each layer, and finally forms 64 small inlets to introduce the sample into the main chamber. Similar bifurcated structures are also applied to the outlet.

For the fabrication of the μCPC device starting from Figure 2a, first, a layer of AZ4620 photoresist was spin-coated on the acetone-cleaned and plasma-treated standard 4” silicon wafer. Then the wafer was UV exposed for 35 s, and was developed in tetramethylammonium hydroxide (TMAH) developer for 90 s as shown in Figure 2b. Then, the micropillars in the μCPC channel were defined by deep reactive ion etching (DRIE) at the rate of 1 μm/min, 10 μm per etching cycle with the maximum column height etched according to the process conditions (the maximum depth to width ratio of 20:1) (Figure 2c). After washing (Figure 2d), the wafer was diced into individual die using a conventional diamond blade dicing saw (HW NPM NANO320). Fixed dicing lines between the active areas were drawn when the layout was designed to avoid silicon debris during the dicing process. In the next step, a layer of 50 nm Ag was sputtered on the surface of the silicon wafer in the vacuum sputtering coater. The surface modification of the micropillars was performed by directly injecting 0.1 mmol/L ADT in dichloromethane to the μCPC channel, then followed by evaporation of the dichloromethane in a vacuum oven. Subsequently, a PDMS cover chip was prepared by the following steps: the PDMS prepolymer and curing agent were mixed with a 10:1 ratio (*v*/*v*) and cured on the patterned metal mold, thoroughly degassed in vacuum, and cured to form a groove corresponding to the size of the silicon wafer. After placing the individual device in the groove, oxygen plasma treatment (PDC-002) of the PDMS and the glass was processed to achieve bonding. Finally, the inlet and outlet of the preconcentrator were perforated and connected with capillary tubes.

### 2.3. Chemical Surface Modification

Figure 3 shows the schematic illustration of surface modification and oximation reaction with carbonyl compounds. ADT was synthesized in accordance with previous research which describes a general method for the immobilization of gold nanoparticles onto solid supports [37]. ADT is immobilized on the surface of the chip through a coordinate covalent bond between Ag and the thiol group. The chemical bonding process was self-assembled at room temperature. The amino-oxy group of ADT can specifically capture carbonyl compounds via the oximation reaction. In the experiment, 0.1 mmol/L of ADT in dichloromethane solution was injected on the surface of the chip, followed by evaporation of dichloromethane on a micro-vibration platform.

### 2.4. Fluorescent Reagent Experiment

To verify the feasibility and detection sensitivity of the protocol, an aldehyde sample with a fluorescent group, 3-perylenecarboxaldehyde, was selected as a typical representative of carbonyl compounds to complete the experiment. The capture capability of the designed device can be qualitatively determined by the change of the fluorescence intensity before and after. Then, qualitative data were obtained through competitive capture experiments of fluorescent aldehydes and other aldehydes. 

The unpacked silicon chip was used to verify whether the carbonyl compound could be detected on the chemically modified surface. The 3-perylenecarboxaldehyde solution diluted to 0.1 μmol/L was selected as a test sample and divided into three sets of experiments. For the first set, the surface of the chip without Ag sputtering was added with 0.1 mmol/L ADT solution and 0.1 μmol/L fluorescent 3-perylenecarboxaldehyde were directly added on the chip sequentially. Then, the chip was eluted with dichloromethane solution. After each step, the sample on the micropillars of the chip was observed by an upright fluorescence microscope (CX41 OLYMPUS) with an excitation wavelength range of 460–490 nm. As 3-perylenecarboxaldehyde is a fluorescent substance it emits fluorescence after being exposed to ultraviolet light. A charge coupled device (CCD) camera was connected above the microscope to export images in the field of view. A 10× objective lens and a 10× eye lens equipped with 1× camera interface adapter were used in the experiment.

In the second set of experiments, the fluorescent sample was added onto the Ag sputtered chip without modification of ADT. Fluorescence images were also taken after the addition of the sample and after the elution. In the third set of experiments, both 0.1 mmol/L of ADT and the fluorescent sample were added onto the Ag sputtered chip. The fluorescent images were also taken after each step of the operation.

After the qualitative verification on the unpacked devices, quantitative tests were also performed on the packed device under microfluidic conditions. In this case, the chemical modified micropillar chips were bonded with the PDMS cover chip. To study the effect of fluid speeds on the capture efficiency, a constant volume (1 mL) of 3-perylenecarboxaldehyde fluorescent solution through a capillary tube was injected into the packed device by a syringe pump. The catheter of the inlet was inserted into the injection needle of the syringe pump, and the flow rate of the injection could be directly input on the syringe pump. The outlet conduits were collected by tubes. The collected samples were dropped on a polished silicon wafer with a pipette to complete the fluorescence detection. The background fluorescence of the wafer, the fluorescence of the original sample, and the fluorescence of the collected solution were recorded and analyzed to study the effect of flow rate on the capture efficiency. After the device had been packaged, it was inconvenient to observe the fluorescence above the micropillars directly using the upright microscope when the device was connecting with capillaries. The effluent liquor was removed directly from the collection tube to a flat silicon wafer by a pipette of 50 μL. The collected samples were dropped onto a smooth silicon wafer with a pipette to complete the fluorescence detection. Each experiment was injected with 3-perylenecarboxaldehyde solution with a fixed concentration of 0.01 μmol/L (2.8 ppb) by a syringe pump. 

## 3. Results and Discussion

### 3.1. Finite Element Simulation

Traditional μCPC consists of tubing packed with granular adsorbent. Packing these tubes with small particles can result in significant pressure drops and sample aggregation, but their size also limits the sampling speed. Wall-coated PCs [38] consist of tubing in which the inner wall is coated with an adsorbent material. They overcome some of the disadvantages of packed PCs but their sample capacity is limited due to their small adsorption surface area. This limitation becomes most dominant in wall-coated μ-PCs due to their small dimensions. One way to address the challenge of increasing surface area without obstructing the flow in a μ-PC is to fabricate micro columns within the μ-PC structure [39]. This was applied in this study to realize the miniaturization of the detection by using the process conditions.

The micro columns are designed in a novel staggered long strip arrangement. The geometric dimensions are shown in Figure 4a. The length of the strip is *W*_1_ = 50 μm and the width is *W*_2_ = 20 μm. The parameter *L*_1_ for measuring the distance between the center points of the strip is used as a variable index in the FEM simulation under COMSOL 5.0 software, where *m* represents the minimum line width between micropillars calculated from the geometric relationship with fixed parameters of length and width of the micro pillars [Appendix A
Figure A1]. This is an important parameter in practical fabrication process.

Two parameters are of importance for the optimum geometry of a preconcentrator device with given external conditions: surface contact area and average fluid velocity. The staggered long strip-shaped micropillar arrangement has more effective reaction area compared to ordinary cylindrical micro columns [28]. For the average fluid velocity, the data of FEM simulation can guide the layout design of μCPC. The average flow velocity of the fluid between the micropillars is simulated under micro-column spacing which is based on the parameter *L*_1_ as the dependent variable ranging from 45–65 microns. As the spacing of the micro columns increases, the average flow velocity of the fluid increases. This means that the μCPC with a small spacing has a longer reaction time, which is preferred for a better capture efficiency. However, on the other hand, when the distance between micropillars is too small, dead zones of fluid flow will exist in the gap of the micropillars in the condition of fixed inlet pressure. To realize the fluid flow, increasing the inlet pressure is required, which will affect the packaging reliability of the device. Therefore, it is necessary to compromise the flow rate and process conditions in order to select the appropriate spacing parameter *L*_1_.

During the simulation process, it was found that the side wall structure on both sides of the flow channel has a large influence on the average flow velocity in the case of low speed flow. The velocity near the side wall of the channel is obviously higher than that inside the μCPC from Figure 4b and upper Figure 4c, thus affecting the capture of subsequent molecules. Therefore, a patterned wall corresponding to the arrangement of the inner long strip micro columns is proposed. Figure 4c shows that the straight wall and the patterned wall cause different flow velocity distribution under the same minimum line width of the device. The straight wall has no flow disturbance to the horizontal flow, and the flow velocity near the straight wall is larger compared with that near the micropillars inside of the channel. It affects the uniform distribution of the overall flow velocity [Appendix A
Figure A2]. However, the patterned wall interferes with the fluid flowing in the horizontal direction, so the flow velocity distribution near the patterned wall is consistent with the flow velocity distribution around the micropillars in the channel, which reduces the ineffective fluid flowing of both sides. 

It can also be seen from the Figure 5 that the design of the patterned wall significantly reduces the average flow velocity of the fluid when the distance between the micro columns is less than 50 μm. From the simulated flow velocity distribution diagram, when the distance between the micro columns is small, the side walls on both sides have a greater influence on the overall flow velocity. After the distance is increased, the influence of the sidewalls is weakened, and the flow velocity between the micro columns plays a leading role.

With the microfabrication facilities used in this work, the highest aspect ratio (pillar height: pillar diameter) is about 10:1; Meanwhile, in order to achieve a better enrichment effect, the micropillars need to have a height more than 150 µm [17]. So the minimum line width *m* allowed for devices is 15 μm. Given the parameters of *W*_1_ = 50 μm, *W*_2_ = 20 μm in the simulation model as shown in Figure 4a, the minimum column spacing *L*_1_ was calculated to be 50 μm. In the simulation, the velocity of fluid rises with increase of *L*_1_, which means that a smaller *L*_1_ can achieve a higher capture efficiency. So the final device design adopts the structure of the patterned wall and the micro column spacing *L*_1_ = 50 μm by considering both the simulation data and the microfabrication conditions. 

For the structural design of inlet and outlet, the simulation results with microchannels in bifurcated design [40] can guide the design of the μCPC, so we adopted bifurcated inlet and outlet channels to promote a uniform distribution of fluid [17].

### 3.2. Device Characterization

The final device package structure is shown in Figure 6e. In order to verify the reliability of the device package, manual injection of colored reagents by capillary was implemented. Figure 6e shows that the flow area of the colored reagent is exactly above the silicon device. This verified that there is a good match with silicon and glass substrates without liquid leakage. Figure 6d shows the easy connection of the system with an injection pump connecting the capillary to inject the sample over the device, while the outlet flows through the capillary to the collection tube. 

The long strip-shaped micro pillars are well-formed and the sides of the micro pillars are steep and flat, as shown in Figure 6b. For circular etching of the micro pillars an etching depth of 184 μm was achieved as shown in Figure 6c. SEM photographs were obtained under the parameters of the SEM: 15 KV accelerating voltage, 2000 times magnification of Figure 6b, and 1000 times magnification of Figure 6c. There were a number of 10^4^ micro pillars in the μCPC, and fluid injection and outflow were realized under the auxiliary of the inlet and outlet PDMS channels on both sides of the silicon chip. Bifurcated channels were adopted at the sample inlet and outlet with the split layer divided into six levels, as shown in Figure 6a. The typical packaging method of silicon microfluidic chips is to achieve silicon and glass bonding through an electrostatic bonding process, but it is difficult to carry out this process after the wafer has been diced for chemical modification, so it is necessary to consider packaging the silicon wafer in the cavity structure of PDMS and realize the closed fluid channel under plasma bonding between PDMS and glass.

### 3.3. Measurements of Fluorescent Aldehyde Capture 

Fluorescence plots for the three sets of unpackaged chips are shown in Figure 7: the chip-controlled variables for each step of surface modification. It can be seen from Figure 7a that the surface of the chip was restored to the original state after the elution of the 3-perylenecarboxaldehyde fluorescent carbonyl compound, indicating that the surface of the chip did not capture the carbonyl compound. In Figure 7b, the surface of the second group of chips was sputtered with Ag but ADT was not added to achieve chemical modification. The surface fluorescence of the chip was also restored to its original state after elution of 3-perylenecarboxaldehyde fluorescent carbonyl compound, indicating that the carbonyl compound could not be captured without chemical modification of ADT. The third set of experiments completed all the surface modification steps including both sputtering of Ag and addition of ADT modification. After elution, there was a large area of high-intensity fluorescence signal on the surface compared with the original background fluorescence, shown in Figure 7c. The fluorescence above the micropillars was observed under the microscope while the modified part was at the side of the micropillars, so focusing the top images was consistent with the side wall images under dripping operation. Another phenomenon that needs to be explained is that no large-area fluorescence was observed above the micropillars after the tested fluorescent sample was added dropwise, but it was observed after elution. The reason is that most of the fluorescent molecules were distributed on the walls of micropillars and they moved from the side to the top of the micropillars after elution due to the scour of fluid. The purpose of the three sets of experiments was to show that the Ag particles can effectively immobilize the ADT molecules, that the ADT molecules can effectively capture the carbonyl compound and that the capture ability proved to be strong. The results verify the feasibility of capturing carbonyl compounds by surface chemistry based on click chemistry.

Figure 7d shows the quantitative results of fluorescence intensity of each set of validation experiment. The fluorescent image is subjected to gradation processing, and the fluorescence intensity value of the surface is extracted by image processing software. Each group of experiments was repeated three times, and each picture took the average fluorescence intensity of several groups of regions, and finally a comparison chart of fluorescence intensity values of micropillars with errors was obtained. The results of the chip function verification can be visualized from the histogram comparison.

Comparison of the three sets of validation experiments results demonstrates that each step of the chemical modification is necessary. The modified surface can capture carbonyl compounds and provide a good platform for subsequent qualitative detection. The limit of detection can be reduced by controlling the volume of the incoming sample after packaging in subsequent experiments. The experimental reaction process takes a short time, and it can be realized at room temperature, which can meet the needs of in-the-field detection.

Figure 8 shows the measured fluorescence intensity of the outlet fluid under the injection of inlet flow rate ranging from 0.1 to 2.5 mL/min. The results clearly show that as the flow rate increases, the fluorescence intensity of the effluent increases. This means that the amount of fluorescent carbonyl compound in the captured sample showed a decreasing trend. The faster the flow rate, the fewer are the molecules captured. The data provides a technical reserve for practical application operations. Figure 8 also shows the comparison between the fluorescence intensity of the sample flowing out at different flow rates with the original sample fluorescence and the original fluorescence of the test platform. It was verified that the packaged device can achieve ppb level (0.01 μmol/L) trace carbonyl compound capture.

Previous experiments used fluorescent aldehyde as an indicator to verify the capture ability of surface-modified μCPC. On this basis, we added acetone solution as a typical carbonyl compound sample by setting a series of concentration ratios between acetone and fluorescent aldehyde (3-perylenecarboxaldehyde) flowing into the device. Since the molecular structure of acetone is much simpler than 3-perylenecarboxaldehyde, the steric hindrance of the reaction between acetone and the amino-oxy group is smaller than 3-perylenecarboxaldehyde. The reaction of acetone occupied the dominant position in the reaction competition, which provided a convenient method for quantitative detection of carbonyl compounds. To study the performance of quantitative analysis, a group of mixed solutions of 3-perylenecarboxaldehyde with concentration of 0.01 μmol/L and acetone with concentration increased from 0.02 μmol/L to 0.07 μmol/L were prepared. Then equal amounts of the mixed solutions (1 mL) were placed in microsamplers fixed on a syringe pump individually and accessed the inlet of device by loading a constant amount of 0.01 nmol ADT via a capillary tube. A set of mixed solutions flew through the device at a flow rate of 1 mL/min. Finally, we measured the fluorescence intensity of the device surface directly. The fluorescence intensity above the micropillars corresponds to the number of molecules captured on the surface of the micropillars. As shown in Figure 9, the fluorescence intensity gradually decreased as a linear function with the correlation coefficient *R*^2^ = 0.9709, 0.9487, 0.9646 corresponding to flow rates of 0.1 mL/min, 1.0 mL/min, 2.5 mL/min when the increased acetone was added. The competitive experiments of non-fluorescent carbonyl compounds and fluorescent carbonyl compounds show that it is possible to measure the concentration of non-fluorescent aldehyde from the fluorescence intensity, from the linear correlation between the fluorescence intensity and concentration of non-fluorescent carbonyl compounds under constant fluorescent carbonyl compound concentration. The obtained results show that our proposed device might provide a feasible way for the detection of trace level carbonyl compounds. [41,42].

## 4. Conclusions

The design, fabrication, and evaluation of a novel microfluidic chip with self-assembled chemically modified surface for capture and analysis of carbonyl compound contaminants in water were described. The results indicate that the device has the potential to concentrate and quantitatively detect carbonyl compounds at the level of ppb. It also provides a rapid and convenient means for in-the-field detection. The proposed columnar preconcentrator structure increases the contact area between the sample and the device surface so that the reaction is sufficient. The procedure of surface modification based on click chemistry is simple without by-products. The oximation reaction between the functional thiol amino-oxy compounds coated on the surfaces of the micropillars and the liquid carbonyl species is the key for the selective capture of the carbonyl compound. The proposed technique makes the present preconcentrator approach attractive not only for chemical carbonyl compounds detection in liquid, but also in gas. 

## Figures and Tables

**Figure 1 sensors-18-04402-f001:**
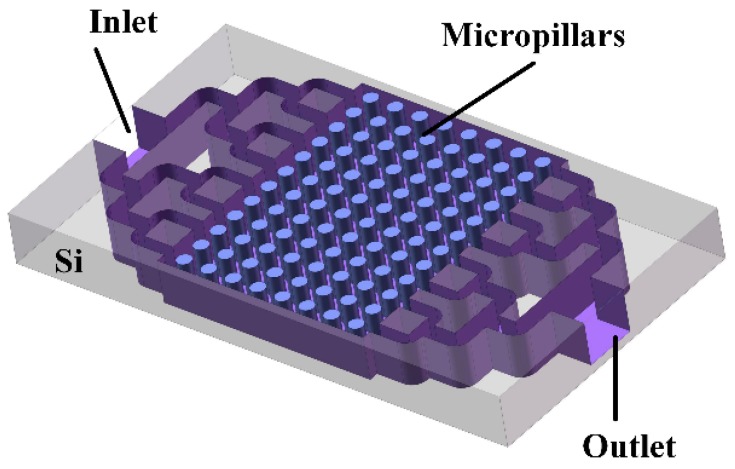
Schematic diagram of the micro columnar preconcentrator (μCPC) (not to scale).

**Figure 2 sensors-18-04402-f002:**
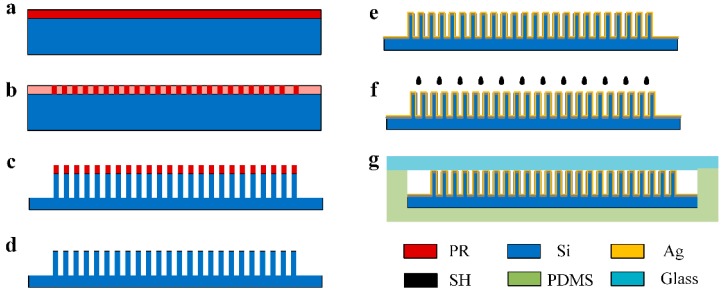
Fabrication process flow for microelectromechanical systems (MEMS)-based μCPC in cross-sectional view. (**a**) Spinning of photoresist (AZ4620); (**b**) Photolithography; (**c**) Deep reactive-ion etching (DRIE); (**d**) Removing glue and plasma treatment; (**e**) Plasma enhanced chemical vapor deposition (PECVD) oxidation followed by deposition of the thin-film precious metal; (**f**) Dispensing of amino-oxy dodecane thiol (ADT) droplets for chemical modification; (**g**) Plasma Bonding.

**Figure 3 sensors-18-04402-f003:**
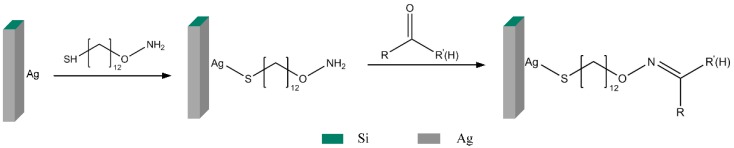
Chemical modification of μCPC surface and trapping of carbonyl compounds via the oximation reaction.

**Figure 4 sensors-18-04402-f004:**
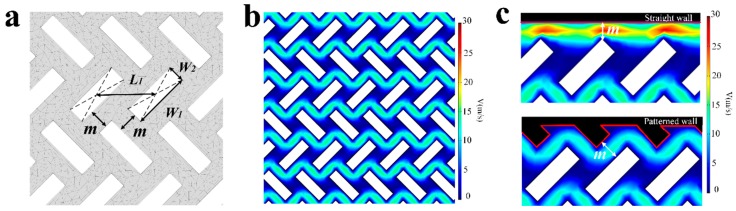
Design of micropillars structure and finite element (FEM) simulation of fluid in the channel. (**a**) Shape and distribution of micropillars; (**b**) Velocity magnitude and flow pattern gained from COMSOL 5.0 software; (**c**) Comparison of simulation results between straight wall and patterned wall under the same minimum line width *m*.

**Figure 5 sensors-18-04402-f005:**
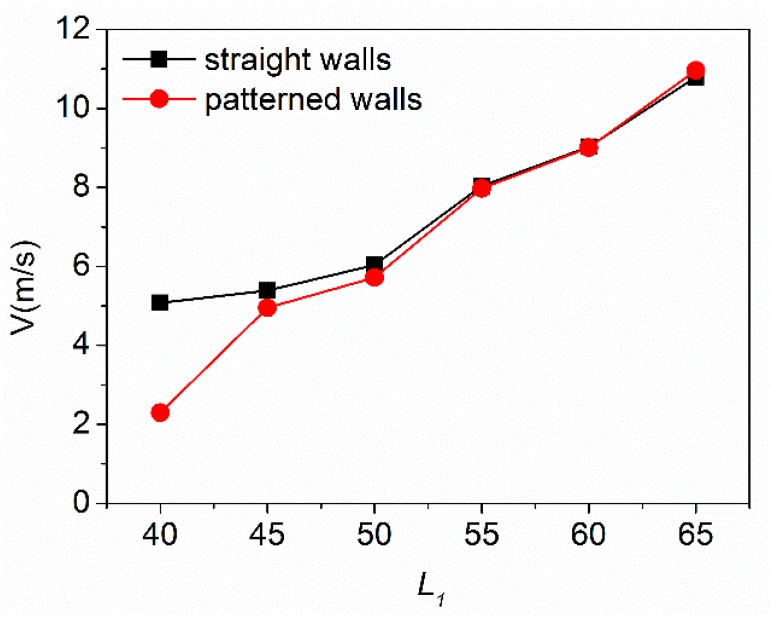
Average flow rate of fluid at different micropillars spacing. Patterned walls show better preconcentrated effect than straight wall under the appropriate flow rate range.

**Figure 6 sensors-18-04402-f006:**
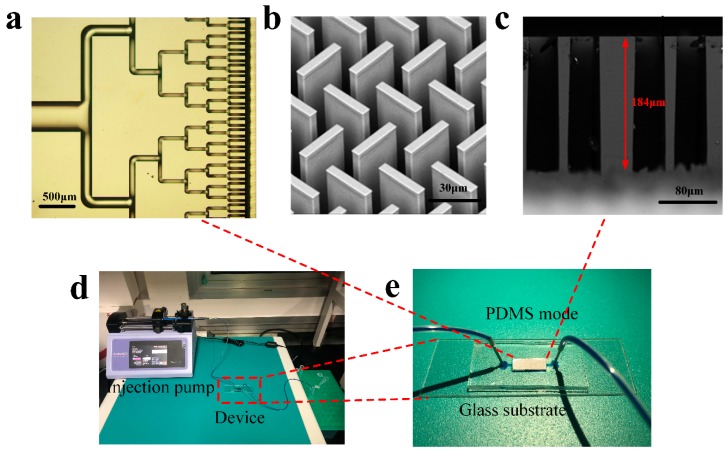
Partial view of the μCPC device. (**a**) Bright field micrograph of bifurcated inlet; (**b**) SEM photograph of micropillars distribution; (**c**) SEM micrograph of the micropillars in cross-sectional view; (**d**) The connection of the system (**e**) Photography image of the packaged structure with color reagent flow through the device.

**Figure 7 sensors-18-04402-f007:**
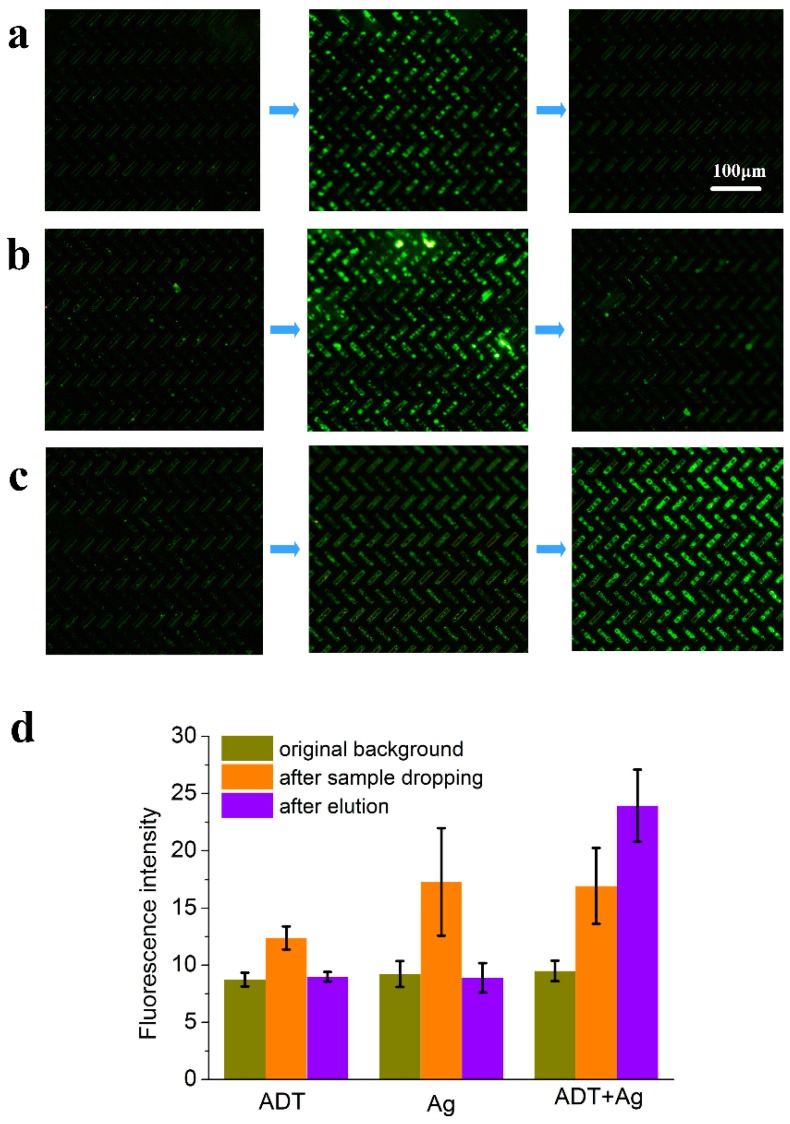
Results of three sets of fluorescence verification experiments. The device of three sets of experiments with different surface modification and surface fluorescence intensity was measured before packaging by the method of dropping (original background, dropping 3-perylenecarboxaldehyde and dichloromethane elution). (**a**) Surface of the micropillars only covered with ADT; (**b**) Surface of the micropillars only covered with precious metal Ag; (**c**) Surface of the micropillars covered with ADT and Ag; (**d**) Fluorescence intensity comparison chart of microfluidic device corresponding to the three sets of experiment.

**Figure 8 sensors-18-04402-f008:**
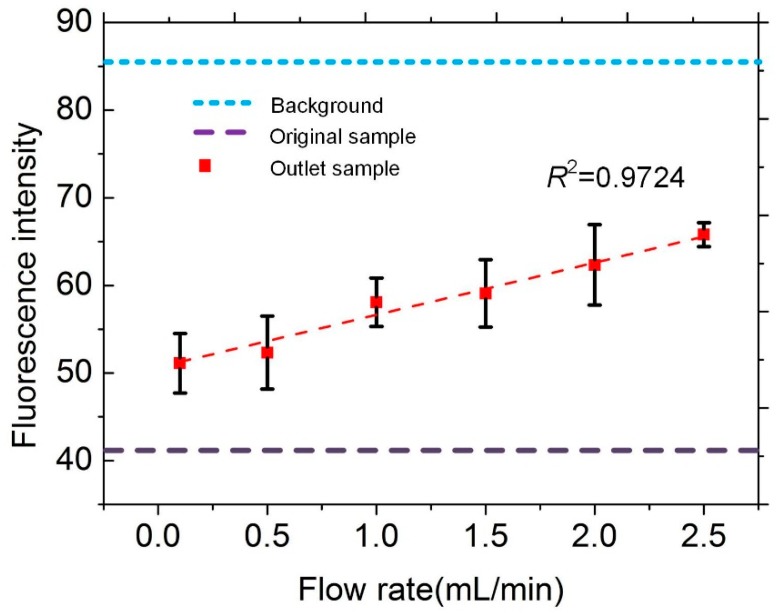
Measured fluorescence intensity of the outlet sample under the injection of inlet flow rate ranging from 0.1 to 2.5 mL/min compared with average fluorescent intensity of the original sample liquid and the background of the packaged microfluidic device.

**Figure 9 sensors-18-04402-f009:**
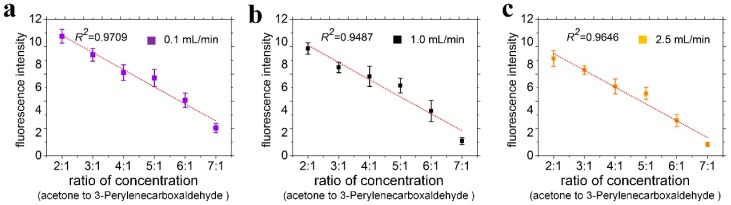
The measured fluorescence intensity and its fitting line under different ratios of acetone and fluorescent aldehyde under flow rates of (**a**) 0.1 mL/min; (**b**) 1.0 mL/min; (**c**) 2.5 mL/min.

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
