# Peer review of "Microfluidic Preconcentration Chip with Self-Assembled Chemical Modified Surface for Trace Carbonyl Compounds Detection"

_sensors, 2018, doi:10.3390/s18124402_

Reviewer 1 Report

In this paper, the authors describe a microfluidic system for capturing carbonyl-containing molecules for preconcentration and analysis of environmental water. To achieve this they describe the optimisation of the fluid dynamics and the functionalisation of a microstructured column-based chip and then test it by exposing it to a fluorescent carbonyl molecule at different concentrations and flow rates.

While their approach is interesting, the paper suffers badly from poor writing, with many sentences difficult to understand (e.g. in the abstract: “While, the demands of miniaturization, simple operation, high sensitivity, and real-time make it challenging for detection of trace level concentration analytes under modern large-scale analytical tools”). The poor writing contributes to, but not wholly accounts for, a lack of clear explanation and several unsubstantiated claims (described in more detail below). As such I recommend this paper is rejected.

Examples of points lacking full explanation:

·         In the FEM experimentation the authors say they simulated structures with straight and convex walls, however the convex walls don’t appear to be shown. I can’t make out any convex walls in Fig. 4c, despite them supposedly being there. The convex walls need to be explicitly shown and the differences relative to the straight wall structures described.

·         The reason for the very different flow patterns in Fig. 4c is not explained beyond saying that one is due to straight wall and the other is due to convex wall. More detail is required.

·         After the FEM simulations, the authors select a column spacing of 50 µm but it is not clear from the accompanying text exactly why that should be the optimum conditions.

·         Figure 8b is explained on page 9 and in its accompanying figure caption, however it is not clear to me exactly what Figure 8b is showing.

·         The experimental is lacking in all details of the electron microscope, optical microscope, and wafer-dicing procedure.

·         The captions to Figures 8b and 9 should describe the flow conditions used in each case.

Examples of unsubstantiated claims:

·         When describing the results shown in Figure 8a, the authors state: “The chemical reaction plays a leading role when the flow rate ranges from 0~0.5 mL/min. The capture of carbonyl molecules are less affected by the flow rate under the low flow rate situation.” Given the large error bars, the authors need to be more circumspect in their claims. The data tentatively suggests that increasing flow rate gives increased fluorescence but, considering the error bars, it’s quite a bold claim to say the data shows that flow rates<0.5 have constant fluorescence.

·         At the bottom of page 9 the authors state “It is verified that the packaged device can achieve ppb level trace carbonyl compound capture.” This might be true, but the authors need to give more details on the calculation they used to arrive at the “ppb” concentration.

·         On page 11 the authors state “The results shows that the device has the potential to quantitatively detect ppb level carbonyl compounds by measuring the fluorescence intensity and provides a means for rapid on-field detection of carbonyl compounds.” This needs justification. Given that the system described here can only measure fluorescent compounds, is it truly applicable to carbonyl compounds generally? How will non-fluorescent compounds be quantified? Moreover, is a system requiring a laboratory fluorescent microscope truly suitable for in-the-field measurements. [As an aside, the correct phrase is “in-the-field”, not “on-field”. This mistake is made throughout the manuscript].

Author Response

Dear Reviewers,

Hereby we would like to submit the revised manuscript entitled “Microfluidic preconcentration chip with self-assembled chemical modificated surface for trace carbonyl compounds detection (ID: sensors-389213)” for consideration for publishing. The authors are Jie Cheng, Jianwei Shao, Yifei Ye, Yang Zhao, Chengjun Huang *, Li Wang * and Mingxiao Li *.

We are very grateful for the reviewers’ constructive comments to improve the quality of the manuscript. We have carefully revised the manuscript according to reviewers’ comments. The revised words and sections were highlighted in the revised manuscript. Meanwhile, a detailed, point-by-point response to the Reviewer’s comments is given below:

Response to Reviewer 1 Comments

In this paper, the authors describe a microfluidic system for capturing carbonyl-containing molecules for preconcentration and analysis of environmental water. To achieve this they describe the optimisation of the fluid dynamics and the functionalisation of a microstructured column-based chip and then test it by exposing it to a fluorescent carbonyl molecule at different concentrations and flow rates.

While their approach is interesting, the paper suffers badly from poor writing, with many sentences difficult to understand (e.g. in the abstract: “While, the demands of miniaturization, simple operation, high sensitivity, and real-time make it challenging for detection of trace level concentration analytes under modern large-scale analytical tools”). The poor writing contributes to, but not wholly accounts for, a lack of clear explanation and several unsubstantiated claims (described in more detail below). As such I recommend this paper is rejected.

Response: We appreciated the reviewer’s interests in our approach, and we apologize for the difficulties in reading the manuscript. In the revised version, we have revised the whole manuscript carefully and tried to avoid any grammar syntax error or misunderstanding. In addition, we also invited our colleagues who are skilled in academic English writing to further improve the language. We hope that the language in the revised manuscript is acceptable for the next review process. We also provided more detailed explanations and reasonable discussion for the results of the simulations and experiments. All revisions and additional information are highlighted in the revised manuscript.

Examples of points lacking full explanation:

      In the FEM experimentation the authors say they simulated structures with straight and convex walls, however the convex walls don’t appear to be shown. I can’t make out any convex walls in Fig. 4c, despite them supposedly being there. The convex walls need to be explicitly shown and the differences relative to the straight wall structures described.

Response: We thank for the good comments. We revised Figure 4c and marked the straight and convex walls of simulated structures on the figures. The straight wall is the side wall of an ordinary microchannel, while the convex walls is a structure corresponding to the shape of the inner micropillars in the inner side wall of the channel.

      The reason for the very different flow patterns in Fig. 4c is not explained beyond saying that one is due to straight wall and the other is due to convex wall. More detail is required.

Response: Figure 4c shows that the straight wall and the convex wall cause different flow velocity distribution due to the shape. We added a short paragraph (Line 222) to explain the flow distribution effects on the different shape of the wall. The straight wall has no flow disturbance to the horizontal flow, so the flow velocity near the straight wall is larger compared with that near the micropillars inside of the channel. It further affects the capture of molecules on the surface of the micropillars because it is easy for fluid to flow away from the vicinity of the straight wall of the channel. However, the convex wall interferes with the fluid flowing in the horizontal direction, so the flow velocity distribution near the wall of the convex groove is consistent with the flow velocity distribution around the micropillars in the channel, which reduces the ineffective fluid flowing of both sides.

      After the FEM simulations, the authors select a column spacing of 50 µm but it is not clear from the accompanying text exactly why that should be the optimum conditions.

Response: From the simulation result in figure 5, we can find the smaller the spacing, the better preconcentration effect achieved. However, from the processing point of view, it is more difficult to fabricate a micropillary array with very small spacing (e.g.,<50µm). With the microfabrication facilities used in this work, the highest aspect ratio (pillar height: pillar diameter) is about 10: 1; Meanwhile, in order to achieve better enrichment effect, the micropillars need to have the height more than 150µm height [17]. Considering these two factors, we chose a minimum line width of 15µm, which corresponds to a column spacing of 50 µm. We revised the description at line (236-238) to explain why a column spacing of 50 µm was chosen in this study.

      Figure 8b is explained on page 9 and in its accompanying figure caption, however it is not clear to me exactly what Figure 8b is showing.

Response: We are sorry for that Figure 8b is confusing. In Figure 8b, our original purpose is to verify that the carbonyl compounds were captured by the chips since the average fluorescence intensity of the outlet sample decreased. In the revised manuscript, we removed Figure 8b and revised Figure 8a by adding the background signal and outlet sample signal on the same figure. The revised Figure 8 shows the measured fluorescence intensity of the outlet sample under the injection of inlet flow rate ranged from 0.1 to 2.5mL/min compared with average fluorescent intensity of the original sample liquid and the background of the packaged microfluidic device. This new figure contains the information of Figure 8a and Figure 8b of the original manuscript, and we think it can verifiy the device's ability of capture fluorescent carbonyl compounds from intuitive comparison in a more clear way.

      The experimental is lacking in all details of the electron microscope, optical microscope, and wafer-dicing procedure.

Response: We thank the good comments. In the revised version, we added the model of the scanning electron microscope (SEM) on Line 85. The SEM was used only when the micropillars were characterized. The parameter of the SEM is 2000 times magnification of Figure 6b, 1000 times magnification of Figure 6c and 15KV accelerating voltage.

We added a short paragraph (Line 163) to explain the details of optical microscope procedure. The microscope used in the experiment is an upright fluorescence microscope (CX41 OLYMPUS).

We used dicing machine with diamond blade to dice the silicon wafer. We added a short paragraph (Line 121) to explain the details of dicing procedure. 

      The captions to Figures 8b and 9 should describe the flow conditions used in each case.

Response: As explained in comment 5, in order to explain Figure 8b clearly, the data from Figures 8a and Figures 8b are combined and a new Figures 8 was added in the revised version. From Figures 8 we can see that the carbonyl compounds trapped on the device surface are reduced as the flow rate increases. In Figure 9, we added two more groups data of experimental data at different flow rates. Three typical flow rate were selected: 0.1 mL/min, 1.0 mL/min and 2.5 mL/min.

Examples of unsubstantiated claims:

      When describing the results shown in Figure 8a, the authors state: “The chemical reaction plays a leading role when the flow rate ranges from 0~0.5 mL/min. The capture of carbonyl molecules are less affected by the flow rate under the low flow rate situation.” Given the large error bars, the authors need to be more circumspect in their claims. The data tentatively suggests that increasing flow rate gives increased fluorescence but, considering the error bars, it’s quite a bold claim to say the data shows that flow rates<0.5 have constant fluorescence.

Response: We appreciate the good comments. Indeed, in the original manuscript, the error bars in Figure 8a are relatively large, and the claims were not very circumspect.

In the revised manuscript, we used a new batch of the devices, and optimized the experimental procedure. In the previous experiments, ADT solution was directly added onto the chip surface with a pipette and the dichloromethane was evaporating in a vacuum oven. This procedure usually resulted in a non-uniform ADT deposition on the chip surface, and a large error bar. In revised manuscript, in order to reduce the error bars, the chip was placed on a micro-vibration platform during the ADT addition and the dichloromethane evaporation, such that the fluorescent signal have more uniform distribution. The obtained results with optimized experimental procedure was plotted as Figure 8 in the revised manuscript.

We added a sentence (Line 145) to explain the optimized experimental procedure and revised our claims on Line 313-320.

      At the bottom of page 9 the authors state “It is verified that the packaged device can achieve ppb level trace carbonyl compound capture.” This might be true, but the authors need to give more details on the calculation they used to arrive at the “ppb” concentration.

Response: The concentration of the 3-perylenecarboxaldehyde solution in this set of experiments was 0.01 μmol/L, it can be converted into 2.8ppb (2.8μg/L) since 3-perylenecarboxaldehyde has a molecular weight of 280(C21H12O). We added the exactly number on Line 189.

      On page 11 the authors state “The results shows that the device has the potential to quantitatively detect ppb level carbonyl compounds by measuring the fluorescence intensity and provides a means for rapid on-field detection of carbonyl compounds.” This needs justification. Given that the system described here can only measure fluorescent compounds, is it truly applicable to carbonyl compounds generally? How will non-fluorescent compounds be quantified? Moreover, is a system requiring a laboratory fluorescent microscope truly suitable for in-the-field measurements. [As an aside, the correct phrase is “in-the-field”, not “on-field”. This mistake is made throughout the manuscript].

Response: We appreciate the reviewer’s very good comments.

Indeed, the statement of “The results shows that the device has the potential to quantitatively detect ppb level carbonyl compounds by measuring the fluorescence intensity and provides a means for rapid on-field detection of carbonyl compounds” was not very circumspect. In this study, we studied the capability of measuring fluorescent compounds on the proposed device. But for the detection of non-fluorescent compounds, we only showed a relatively simple demonstration of non-fluorescent carbonyl compounds detection in a competitive experiment, which indicated one possibility for non-fluorescent compounds detection.

In the revised manuscript, we revised the discussion accordingly at line 342-351, and added two more related reference in the manuscript as below:

[1] Chang J , Pablo Arbeláez, Switz N , et al. Automated tuberculosis diagnosis using fluorescence images from a mobile microscope. International Conference on Medical Image Computing & Computer-assisted Intervention. 2013.DOI: 10.1007/978-3-642-33454-2_43

[2] Breslauer D N , Maamari R N , Switz N A , et al. Mobile Phone Based Clinical Microscopy for Global Health Applications[J]. PLOS ONE, 2009, 4(7):e6320..DOI: 10.1371/journal.pone.0006320

We also changed “on-field” to “in-the-field” throughout the manuscript. 

In case of further information please feel free to contact me. Looking forward to hearing from you.

Sincerely Yours

Dr. Jie Cheng

Institute of Microelectronics of Chinese Academy of Sciences

Reviewer 2 Report

Report Sensors

The introduction is poor in terms of microfluidics example and the reported literature is dated. More recent works and reviews on microfluidics devices and fabrication method should be considered. A list of possible papers is reported below:

Optimization of microfluidic biosensor efficiency by means of fluid flow engineering. Sci. Rep.2017.

Optimization of a suspended two photon polymerized microfluidic filtration system. Microelectron. Eng.2018,195, 95–100.

On-line sample pre-concentration in microfluidic devices: A review. Anal. Chim. Acta2012.

Sample pre-concentration with high enrichment factors at a fixed location in paper-based microfluidic devices. Lab Chip2016.

At line 59. Revise “a new means”

About the simulation

The Author report about the simulation with simulation software COMSOL, but there is not a reference math model.

About the process

First

The presented fabrication chip process use different materials: Silicon, PDMS, glass. It is not so clear why you need both glass and PDMS. Why don’t you use only glass or only PDMS for the chip cover?

Second

In the inlet/outlet ports are on the PDMS, how you can ensure a good match with silicon and glass substrates without liquids leakage? This should be explained and better showed in figure 6a. A new image that clarify the connection of the different parts of the system should be provided.

Author Response

Dear Reviewers, Hereby we would like to submit the revised manuscript entitled “Microfluidic preconcentration chip with self-assembled chemical modificated surface for trace carbonyl compounds detection (ID: sensors-389213)” for consideration for publishing. The authors are Jie Cheng, Jianwei Shao, Yifei Ye, Yang Zhao, Chengjun Huang *, Li Wang * and Mingxiao Li *. We are very grateful for the reviewers’ constructive comments to improve the quality of the manuscript. We have carefully revised the manuscript according to reviewers’ comments. The revised words and sections were highlighted in the revised manuscript. Meanwhile, a detailed, point-by-point response to the Reviewer’s comments is given below: Response to Reviewer 2 Comments The introduction is poor in terms of microfluidics example and the reported literature is dated. More recent works and reviews on microfluidics devices and fabrication method should be considered. A list of possible papers is reported below: Optimization of microfluidic biosensor efficiency by means of fluid flow engineering. Sci. Rep.2017. Optimization of a suspended two photon polymerized microfluidic filtration system. Microelectron. Eng.2018,195, 95–100. On-line sample pre-concentration in microfluidic devices: A review. Anal. Chim. Acta2012. Sample pre-concentration with high enrichment factors at a fixed location in paper-based microfluidic devices. Lab Chip2016. Response: More recent works and reviews on microfluidics devices and fabrication method are supplemented in the introduction (Line 60-65) and some outdated literature references are replaced. At line 59. Revise “a new means” Response: “A new means” has been replaced with “new means”. About the simulation The Author report about the simulation with simulation software COMSOL, but there is not a reference math model. Response: We have added a short paragraph (Line 103) to explain the math model. Physics field of simulation select laminar flow. In the microfluidic channel, the flow velocity field is described by the Navier-Stokes equations. The fluid is assumed to be Newtonian and incompressible. The simulation model is in the 2D Cartesian coordinates and for a steady flow. The fluid properties such as the dynamic viscosity and the density are assumed to be constant. The dynamic viscosity is μ = 10−3 Pa.s, and the fluid density is ρ = 1000 kg/m3. The microfluidic channel shape as fluid wall affect flow rate and pressure of the fluid. About the process First The presented fabrication chip process use different materials: Silicon, PDMS, glass. It is not so clear why you need both glass and PDMS. Why don’t you use only glass or only PDMS for the chip cover? Response: The functional part of the presented fabrication chip is silicon material which were formed the micropillars by etching and chips were arranged densely on the 4 inch silicon wafer. In order to avoid contaminating between adjacent devices when performing different chemical modifications, the wafer need to be diced and each device was further individually processed on die level. If only use glass for the chip cover, is difficult to realize bonding between single device and glass. Generally, electrostatic bonding requires complete silicon wafer. The device area is too small to meet the process conditions, so we considered to use PDMS to build a frame and have the aid of the bonding between PDMS and glass to realize the package of device. The reason of not using only glass or only PDMS for the chip cover is bonding between PDMS and glass is easy to implement, and this method reduces the cost of the chip. The bonding between glass and PDMS is more stable than that between PDMS and PDMS. If the glass is used as the frame, the etching of the glass is expensive and difficult to achieve. It is why we finally use PDMS to build a frame of the chip and glass for chip cover. Second In the inlet/outlet ports are on the PDMS, how you can ensure a good match with silicon and glass substrates without liquids leakage? This should be explained and better showed in figure 6a. A new image that clarify the connection of the different parts of the system should be provided. Response: We changed Figure 6 to explain the connection of different parts. We also added more figures in Figure 6 and a short paragraph (Line 246) to show the reliability of the device by manual injection of colored reagents by capillary. Figure 6e shows that the flow area of the colored reagent is above the silicon device exactly. It verified that there is a good match with silicon and glass substrates without liquids leakage. In case of further information please feel free to contact me. Looking forward to hearing from you. Sincerely Yours Dr. Jie Cheng Institute of Microelectronics of Chinese Academy of Sciences E-mail: [email protected]

Round  2

Reviewer 1 Report

The authors have made significant improvements to address a lot of my concerns and I think the paper is much clearer as a consequence. A few concerns still remain however:

1.            Firstly (and most importantly) the extra details and discussion concerning the computational experiments contain seemingly unsubstantiated claims. For example:

“The straight wall has no flow disturbance to the horizontal flow, so the flow velocity near the straight wall is larger compared with that near the micropillars inside of the channel.”

Looking at figure 4c top, this appears to be wrong:  the velocity near the walls seem to peak at ~11 m/s at the narrowest point, whereas the flow near the micropillars inside the channel seem to be ~13 m/s. (As an aside it would be much easier to compare the two figures in 4c if they had the same colour scaling). Also, this is just considering linear velocities. If we were to consider volumetric flow, the flow would be even greater in the centre channel as the flow constrictions are much narrower by the wall.

As a paper should be scientifically sound as a minimum requirement, it is concerning that the data seems to contradict the claims made in the main text.

Earlier the authors state “If the side walls of the flow channel are too far from the micro columns, most of the fluid will flow away from the gaps on both sides and thus affecting the capture of subsequent molecules.” This suggests that they might have performed additional experiments (not shown in the paper) varying the distance of the straight wall from the pillar structures, so maybe the unsubstantiated claim quoted above refers to another experiment where the wall was further from the pillars?

Additionally in line 210 “…and the average flow velocity of the fluid between the micropillars is simulated from 45 to 65 microns.” Microns are not a unit of velocity.

2.            “We revised Figure 4c and marked the straight and convex walls of simulated structures on the figures. The straight wall is the side wall of an ordinary microchannel, while the convex walls is a structure corresponding to the shape of the inner micropillars in the inner side wall of the channel.”

The use of the word “convex” here is misleading and led to my confusion in the original manuscript. I think the authors are using it here in the sense of “extending outward”, however “convex” is usually associated with a curved surface, or possibly a polygon with obtuse angles (a “convex polygon”). Hence in the original manuscript I was looking for a curved wall that wasn’t there! I think “patterned walls” or “patterned surface” would be a more apt description.

3.             Thirdly in my original comments I said “After the FEM simulations, the authors select a column spacing of 50 µm but it is not clear from the accompanying text exactly why that should be the optimum conditions.” Neither the response nor the extra sentence in the manuscript (lines 236-238) make this much clearer. In the response the authors state that “…we chose a minimum line width of 15µm, which corresponds to a column spacing of 50 µm.” it is not clear to me how the 50 µm derives from the 15 µm.

4.            The authors added in the carbonyl concentration in the experimental (line 189) to substantiate the “ppb level trace carbonyl compound capture.” (line 320). It would be useful if the concentration was also quoted alongside that statement in line 320.

Author Response

Dear Reviewers,

Hereby we would like to submit the revised manuscript entitled “Microfluidic preconcentration chip with self-assembled chemical modificated surface for trace carbonyl compounds detection (ID: sensors-389213)” for consideration for publishing. The authors are Jie Cheng, Jianwei Shao, Yifei Ye, Yang Zhao, Chengjun Huang *, Li Wang * and Mingxiao Li *. 

We are very grateful for the reviewers’ constructive comments to improve the quality of the manuscript. We have carefully revised the manuscript according to reviewers’ comments. The revised words and sections were highlighted in the revised manuscript. Meanwhile, a detailed, point-by-point response to the Reviewer’s comments is given below:

Response to Reviewer 1 Comments

•      Firstly (and most importantly) the extra details and discussion concerning the computational experiments contain seemingly unsubstantiated claims. For example:

“The straight wall has no flow disturbance to the horizontal flow, so the flow velocity near the straight wall is larger compared with that near the micropillars inside of the channel.”

Looking at figure 4c top, this appears to be wrong:  the velocity near the walls seem to peak at ~11 m/s at the narrowest point, whereas the flow near the micropillars inside the channel seem to be ~13 m/s. (As an aside it would be much easier to compare the two figures in 4c if they had the same colour scaling). Also, this is just considering linear velocities. If we were to consider volumetric flow, the flow would be even greater in the centre channel as the flow constrictions are much narrower by the wall.

As a paper should be scientifically sound as a minimum requirement, it is concerning that the data seems to contradict the claims made in the main text.

Earlier the authors state “If the side walls of the flow channel are too far from the micro columns, most of the fluid will flow away from the gaps on both sides and thus affecting the capture of subsequent molecules.” This suggests that they might have performed additional experiments (not shown in the paper) varying the distance of the straight wall from the pillar structures, so maybe the unsubstantiated claim quoted above refers to another experiment where the wall was further from the pillars?

Additionally in line 210 “…and the average flow velocity of the fluid between the micropillars is simulated from 45 to 65 microns.” Microns are not a unit of velocity.

Response: We thank for the good comments. Indeed, the model we used in the original simulation was not very rigorous. Namely, the minimum line width was ignored in the simulation for the simplicity reason, which probably led to inconsistency of results in the manuscript. In fact, the distance between the straight wall and the boundary of the closest micro pillars set equals to the minimum line width m. In the revised manuscript, we added a minimum line width in the simulation model, and repeated the simulation. In this case, the flow velocity distribution of the straight wall and the patterned wall at the same minimum line width can be seen in Figure 4c. From the same color scaling, we got the conclusion that the flow velocity near the straight wall is larger compared with that near the micropillars inside of the channel. We revised simulation conditions and modified the simulation diagram of straight wall in Figure 4c.

We reorganized the language to explain why we proposed the patterned walls. The flow velocity in the channel near the sidewall is higher than that of the internally distributed micro pillars, so the straight wall affects the uniform distribution of the overall flow velocity. In addition, in the revised manuscript, we provided an extra simulation result to show the effect of the straight wall on the flow velocity, as shown in the FigureA2 in the supplementary material.

Additionally in line 212, original sentence caused misunderstanding, so we revised the sentence to “The average flow velocity of the fluid between the micropillars is simulated under micro-column spacing which based on the parameter L1 as the dependent variable ranges from 45 to 65 microns.”

•      “We revised Figure 4c and marked the straight and convex walls of simulated structures on the figures. The straight wall is the side wall of an ordinary microchannel, while the convex walls is a structure corresponding to the shape of the inner micropillars in the inner side wall of the channel.”

The use of the word “convex” here is misleading and led to my confusion in the original manuscript. I think the authors are using it here in the sense of “extending outward”, however “convex” is usually associated with a curved surface, or possibly a polygon with obtuse angles (a “convex polygon”). Hence in the original manuscript I was looking for a curved wall that wasn’t there! I think “patterned walls” or “patterned surface” would be a more apt description.

Response: We thank for the good comment and suggestion. In the revised manuscript, we modify the “convex walls” to “patterned walls” in line 206, line 225, line 226, line 230, line 247, etc.

•      Thirdly in my original comments I said “After the FEM simulations, the authors select a column spacing of 50 µm but it is not clear from the accompanying text exactly why that should be the optimum conditions.” Neither the response nor the extra sentence in the manuscript (lines 236-238) make this much clearer. In the response the authors state that “…we chose a minimum line width of 15µm, which corresponds to a column spacing of 50 µm.” it is not clear to me how the 50 µm derives from the 15 µm.

Response: The parameter L1 for measuring the distance between the center points of the strip is used as variable index in the FEM simulation, where m represents the minimum line width between micropillars calculated from the geometric relationship with fixed parameters of the length and width of the micro pillars, which is an important parameter in practical fabrication process. With the microfabrication facilities used in this work, the highest aspect ratio (pillar height: pillar diameter) is about 10: 1; Meanwhile, in order to achieve better enrichment effect, the micropillars need to have the height more than 150µm height [17]. So the minimum line width m allowed for devices is 15μm. Given the parameters of W1=50 μm,W2=20 μm in the simulation model as shown in Figure 4a,the minimum column spacing L1 could be calculated to be 50μm. In the simulation, so the velocity of fluid raises with the increase of L1, which means that a smaller L1 can achieve a higher capture efficiency. Therefore, we selected L1= 50 μm in this study for the subsequent process.

In the revised manuscript, we revised the figure 4a for marking the geometry parameters of micro pillars. Also, we added the explanation for why we select a column spacing of 50 µm in the manuscript. (lines 239-245). Meanwhile, we introduced the simulation model in details, as shown in Figure A1 in the supplementary material.

•      The authors added in the carbonyl concentration in the experimental (line 189) to substantiate the “ppb level trace carbonyl compound capture.” (line 320). It would be useful if the concentration was also quoted alongside that statement in line 320.

Response: We added the concentration (0.01 μmol/L) alongside the statement in line 328 in order to be consistent with the above(line 191).

In case of further information please feel free to contact me. Looking forward to hearing from you.

Sincerely Yours

Dr. Jie Cheng

Institute of Microelectronics of Chinese Academy of Sciences

Reviewer 2 Report

Ok for publication in the present form

Author Response

Dear Reviewers,

Hereby we would like to submit the revised manuscript entitled “Microfluidic preconcentration chip with self-assembled chemical modificated surface for trace carbonyl compounds detection (ID: sensors-389213)” for consideration for publishing. The authors are Jie Cheng, Jianwei Shao, Yifei Ye, Yang Zhao, Chengjun Huang *, Li Wang * and Mingxiao Li *.

We are very grateful for the reviewers’ constructive comments to improve the quality of the manuscript.

Sincerely Yours

Dr. Jie Cheng

Institute of Microelectronics of Chinese Academy of Sciences

Round  3

Reviewer 1 Report

I thank the authors for the improvements.